# Cefiderocol: Systematic Review of Mechanisms of Resistance, Heteroresistance and In Vivo Emergence of Resistance

**DOI:** 10.3390/antibiotics11060723

**Published:** 2022-05-27

**Authors:** Stamatis Karakonstantis, Maria Rousaki, Evangelos I. Kritsotakis

**Affiliations:** 1Internal Medicine Department, Infectious Diseases Division, University Hospital of Heraklion, 71500 Heraklion, Greece; 2Master of Public Health Program, Department of Social Medicine, School of Medicine, University of Crete, 70013 Heraklion, Greece; maria.rou@hotmail.com; 3Laboratory of Biostatistics, Department of Social Medicine, School of Medicine, University of Crete, 70013 Heraklion, Greece; e.kritsotakis@uoc.gr

**Keywords:** cefiderocol, resistance, heteroresistance

## Abstract

Cefiderocol appears promising, as it can overcome most β-lactam resistance mechanisms (including β-lactamases, porin mutations, and efflux pumps). Resistance is uncommon according to large multinational cohorts, including against isolates resistant to carbapenems, ceftazidime/avibactam, ceftolozane/tazobactam, and colistin. However, alarming proportions of resistance have been reported in some recent cohorts (up to 50%). A systematic review was conducted in PubMed and Scopus from inception to May 2022 to review mechanisms of resistance, prevalence of heteroresistance, and in vivo emergence of resistance to cefiderocol during treatment. A variety of mechanisms, typically acting in concert, have been reported to confer resistance to cefiderocol: β-lactamases (especially NDM, KPC and AmpC variants conferring resistance to ceftazidime/avibactam, OXA-427, and PER- and SHV-type ESBLs), porin mutations, and mutations affecting siderophore receptors, efflux pumps, and target (PBP-3) modifications. Coexpression of multiple β-lactamases, often in combination with permeability defects, appears to be the main mechanism of resistance. Heteroresistance is highly prevalent (especially in *A. baumannii*), but its clinical impact is unclear, considering that in vivo emergence of resistance appears to be low in clinical studies. Nevertheless, cases of in vivo emerging cefiderocol resistance are increasingly being reported. Continued surveillance of cefiderocol’s activity is important as this agent is introduced in clinical practice.

## 1. Introduction

Extensively drug-resistant (XDR) and pandrug-resistant Gram-negative bacteria are increasingly being reported worldwide [1]. Attributable mortality appears to be high [2], and treatment options very limited [3], with synergistic combinations of in vitro inactive (alone) antimicrobials often being a last resort [3,4,5]. Cefiderocol appears promising, as it can overcome most of the mechanisms of β-lactam resistance (including β-lactamases, porin mutations and efflux pumps) [3].

Cefiderocol is a novel siderophore cephalosporin [6,7]. It has a structure similar to that of cefepime (pyrrolidinium group on the C-3 side chain, which improves stability against β-lactamases) and ceftazidime (same C-7 side chain conferring improved stability against β-lactamases and improved transport across the outer membrane) [6,7]. The major difference is the addition of a chlorocatechol group on the end of the C-3 side chain, which further enhances β-lactamase stability and confers siderophore activity [6,7]. Siderophores are natural iron-chelating molecules used by bacteria to facilitate iron transport. Notably, the natural siderophores enterobactin (*E. coli*) and pyoverdine (*P. aeruginosa*) contain similar catechol groups as an iron-chelating moiety. By utilizing natural iron transportation systems (often referred to as a “Trojan horse” strategy), cefiderocol is actively transported across the outer membrane of Gram-negative bacteria, therefore overcoming resistance mediated by porin loss or efflux pumps [6]. This, in combination with improved β-lactamase stability, makes cefiderocol an ideal candidate for treatment of infections by XDR/PDR Gram-negative bacteria.

Based on several prior studies, including large multinational cohorts [8,9,10,11,12,13], cefiderocol appears to be active against the majority of Enterobacterales, *A. baumannii*, *P. aeruginosa*, and *S. maltophilia*, including isolates that are resistant to carbapenems, ceftazidime/avibactam, ceftolozane/tazobactam, meropenem/vaborbactam, imipenem/relabactam, and polymyxins. Nevertheless, cefiderocol resistance has been reported to be high in selected cohorts [14,15,16,17,18,19]. The prevalence of heteroresistance may also be high [20], and cases of resistance emerging during treatment are increasingly being reported [21,22,23,24,25,26]. The aim of this manuscript was to systematically review mechanisms of resistance to cefiderocol, prevalence of heteroresistance and reports of in vivo emerging resistance.

## 2. Results

The study flow chart is depicted in Figure 1. A total of n = 52 relevant studies were reviewed [12,14,15,16,17,19,20,21,22,23,24,25,26,27,28,29,30,31,32,33,34,35,36,37,38,39,40,41,42,43,44,45,46,47,48,49,50,51,52,53,54,55,56,57,58,59,60,61,62,63,64]. All studies were published after 2018; n = 17 (33%) in 2022, n = 18 (35%) in 2021, n = 12 (23%) in 2020, n = 1 (2%) in 2019, and n = 4 (8%) in 2018. Evaluation of mechanisms of resistance was based on: n = 13 group 1 studies [14,17,27,29,30,37,45,52,53,54,61,63,64], n = 4 group 2 studies [12,15,19,62], n = 11 group 3 studies [14,16,21,24,25,32,42,43,44,55,59], n = 7 group 4 studies [28,31,40,46,57,58,60], and n = 13 group 5 studies [29,31,41,47,48,49,50,51,54,55,56,57,64] (see Section 4.2 for grouping of studies). Data about the prevalence of heteroresistance were available in only two studies [20,33]. Data about resistance emerging in vivo (during/after cefiderocol treatment) were available in 12 studies [21,22,23,24,25,26,32,34,35,36,38,39].

### 2.1. Role of β-Lactamases

Various β-lactamases have been correlated to cefiderocol resistance, and we discuss the most notable below. A detailed summary of the available data is presented in Table A1.

#### 2.1.1. NDM Metallo-β-Lactamases (MBL)

The role of NDM-type β-lactamases has been supported by several findings, including multifold increases in cefiderocol MIC by introduction of NDM in isogenic mutants [29,54,64] and much higher prevalence (42–59% in some cohorts [12,19,62]) of cefiderocol nonsusceptibility in NDM-producing clinical isolates. Additionally, in vivo emergence of cefiderocol resistance in *E. coli* associated with increased copy numbers of *bla_NDM_* genes has been reported [25]. Moreover, expression of NDM appears to facilitate the emergence of cefiderocol resistance by additional mechanisms (such as mutations in siderophore receptors) [40,64].

#### 2.1.2. KPC Variants

The role of KPC variants, particularly those conferring resistance to ceftazidime/avibactam, has been supported by both multifold increases in MIC by introduction of selected KPC variants in isogenic mutants [41,56,64] and clinical data [12,42,43,56]. Notably, cefiderocol resistance (MIC > 2 mg/L) was considerably higher (83% vs. 7%) in ceftazidime/avibactam-resistant (n = 40) than in ceftazidime/avibactam-susceptible (n = 60) KPC-producing Enterobacterales (mostly *K. pneumoniae*) in one cohort [12]. In addition, in vivo emerging resistance to cefiderocol in *K. pneumoniae* attributable to KPC variants has been reported [12,42,56].

There is evidence that binding of cefiderocol by KPC-3 variants KPC-41 and KPC-50 (rather than hydrolysis) is responsible for reduced susceptibility to cefiderocol [56]. Therefore, lack of hydrolysis does not exclude a contribution of β-lactamase to cefiderocol resistance.

#### 2.1.3. Role of OXA-Type β-Lactamases

Among OXA-type β-lactamases, only OXA-427 has been correlated to cefiderocol resistance [15]. OXA-427 is a novel type of carbapenem-hydrolysing class D β-lactamase, sharing only 22–29% amino acid identity with OXA-48-like enzymes and conferring resistance to extended-spectrum cephalosporins (mostly ceftazidime) [65]. Uniform cefiderocol nonsusceptibility on disk diffusion was reported among n = 26 OXA-427-producing Enterobacterales isolates in Belgium [15]. OXA-427 was been recently confirmed to have hydrolytic activity against cefiderocol [56], although introduction of OXA-427 in *E. coli* resulted in only twofold increase in cefiderocol MIC [56]. Various OXA-type β-lactamases (especially OXA-23) have also been commonly reported in cefiderocol-resistant *A. baumannii* clinical isolates [17,29,30,63]. However, cefiderocol appears to be stable against OXA-23 (as well as OXA-48 and OXA-40) [66]. Furthermore, introduction of OXA-1, OXA-48, OXA-23, OXA-24, OXA-40, OXA-58, and OXA-232 in *E. coli*/*K. pneumoniae*/*A. baumannii*/*P. aeruginosa* have not been shown to affect cefiderocol MIC [29,32,54,56,64,66].

#### 2.1.4. Role of PER-Type, SHV-Type, and BEL-Type ESBLs

Both PER-type [29,54,56] and SHV-type EBSLs [29,54,56] have been associated with increased cefiderocol MIC following their introduction to isogenic mutants. Both types of ESBLs have been detected in cefiderocol-resistant clinical isolates. Specifically, PER-type ESBLs have been detected predominantly in cefiderocol-resistant *A. baumannii* [27,29,54], but also in *P. aeruginosa* [19]. SVH-type β-lactamases have been correlated to cefiderocol resistance in *K. pneumoniae* and *A. baumannii* [29]. BEL-type β-lactamases also have the potential to contribute to cefiderocol resistance based on isogenic mutant experiments [54,56].

#### 2.1.5. Role of AmpC Variants

In vivo emerging cefiderocol cross-resistance attributable to *ampC* mutations was reported in two patients, with *Enterobacter cloacae* infections being treated with ceftazidime/avibactam [43]. The role of these AmpC variants in cefiderocol resistance has also been confirmed by introducing them in isogenic *E. coli* mutants, resulting in a 32-fold increase in cefiderocol MIC by A292_L293del AmpC variant [43] and a 4-fold increase by A294_P295del AmpC [43]. In vivo emerging cefiderocol cross-resistance potentially attributable to *ampC* mutation was reported in three patients with *P. aeruginosa* infections being treated with ceftolozane/tazobactam [16,44].

#### 2.1.6. Reversal of Cefiderocol Susceptibility by β-Lactamase Inhibitors

The role of β-lactamases in cefiderocol resistance has been further supported by potentiation of cefiderocol activity by dipicolinic acid (an MBL inhibitor) against MBL-producing (mainly NDM-producing) isolates and by avibactam against isolates producing serine β-lactamases (e.g., SHV- or PER-type ESBLs) [19,29,30,40,47,63,64]. In isolates coexpressing both MBL and serine β-lactamases, potentiation of cefiderocol activity is greatest in the presence of both dipicolinic acid and avibactam [29].

### 2.2. Permeability Defects/Increased Efflux

Mechanisms affecting siderophore receptors are summarized in Table A2, while mechanisms involving porins and efflux pumps are summarized in Table A3.

#### 2.2.1. Mutations Affecting Siderophore Receptors

Several studies based οn isogenic mutants have proven the role of siderophore receptor mutations in cefiderocol resistance. PiuA and PiuD (an ortholog of PiuA) appear to be most important in *P. aeruginosa* [47,50]. Mutations affecting *pirA* alone appear to be less important in *P. aeruginosa* but may further raise cefiderocol MIC in combination with mutations affecting either *piuA* or *piuD* [50]. One study, however, did not find a correlation between siderophore receptors (including PiuA/PiuD and PirA) and cefiderocol MIC in 10 *P. aeruginosa* isolates with cefiderocol MIC ranging from 0.03 to 8 mg/L [45]. In Enterobacterales, mutations in siderophore receptors CirA and/or Fiu appear to be most important [40,47,57,64], especially in the presence of NDM MBLs [40,57,64]. Differences in the degree of resistance caused by mutations in specific siderophore receptors may be due to the relative contribution of each iron acquisition system in a given strain [57]. Mutations affecting iron transport have also been detected in in vitro derived (after serial passaging) cefiderocol-resistant *S. maltophilia* [58,60]. Mutations affecting the aforementioned genes have also been detected in clinical cefiderocol-resistant isolates, including *P. aeruginosa* (*piuD* and *pirR*) [44], *A. baumannii* (*pirA* and *piuA*) [52], and *K. pneumoniae* (*cirA*) [24].

#### 2.2.2. Porin Mutations

Porin mutations can raise cefiderocol MIC based on isogenic *K. pneumoniae* (OmpK35, OmpK36) and *P. aeruginosa* (OprD) mutants [47]. Mutations in porins have also been detected in cefiderocol-resistant clinical *K. pneumoniae* (OmpK35, OmpK36, OmpK37) and *Enterobacter* spp. (OmpC, OmpF) isolates, in combination with expression of various β-lactamases [37,53,55]. Finally, a high percentage of cefiderocol resistance (38.5% with MIC > 2 mg/L) was reported in ESBL-Enterobacterales with porin mutations [19].

#### 2.2.3. Overexpression of Efflux Pumps

Efflux pumps may contribute to cefiderocol resistance based on transformed isogenic mutants or in vitro derived cefiderocol-resistant mutants in *K. pneumoniae* (sugE and chrA) [37], *P. aeruginosa* (MexAB–OprM efflux pump) [47], and *S. maltophilia* (smeDEF efflux pump) [60]. Furthermore, overexpression of AxyABM efflux pump in *Achromobacter xylosoxidans* was associated with a threefold higher cefiderocol MIC [51].

### 2.3. Target Modification and Other Genes Potentially Involved in Cefiderocol Resistance

Target modification (mutations in PBP-3) has been noted in few cefiderocol-resistant *E. coli* [31,59] and *A. baumannii* [30,52]. However, the role of this mechanism in cefiderocol resistance is uncertain, and based on isogenic *E. coli* mutants, it only minimally raised (2-fold) cefiderocol MIC [31]. Mutations in other genes potentially involved in cefiderocol resistance are summarized in Table A4. Most of these mutations have been detected only in vitro and not in clinical isolates.

### 2.4. Combination of Mechanisms Contribute to Cefiderocol Resistance

Considering the results of studies of isogenic strains, each of the mechanisms described above appears to be insufficient to raise, on its own, cefiderocol MICs above current PK/PD breakpoints [29,31,40,46,47,50,51,54,55]. Indeed, the majority of strains harbouring various mechanisms of resistance, including various β-lactamases, remain susceptible to cefiderocol [8,9,10,11,12,47]. This suggests that a combination of different mechanisms is necessary, including coexpression of different β-lactamases, mutations affecting siderophore–drug receptors expression/function, mutations affecting porin expression/function, overexpression of efflux pumps, and target (PBP-3) modification [37,59,64]. This is further supported by the fact that various studies of clinical isolates have not correlated cefiderocol resistance with specific mechanisms but detected multiple resistance mechanisms comprising predominantly coexpression of various β-lactamases in combination with permeability defects in cefiderocol-resistant isolates [17,37,45,53,63].

Furthermore, a higher baseline cefiderocol MIC due to the presence of one or more of the above mechanisms of resistance appears to facilitate the emergence of cefiderocol resistance. The closer the MIC is to the breakpoints, the easier it is for additional mutations to raise the MIC above PK/PD breakpoints [40,64]. For example, NDM-5 production has been shown to facilitate emergence of cefiderocol resistance (by additional mutations affecting CirA) in *E. cloacae* [40] and *K. pneumoniae* [64]. Notably, the emergence of resistance was not possible in vitro in *E. cloacae* isolates with alternative β-lactamases (not affecting cefiderocol) and was prevented in the presence of dipicolinic acid [40].

Finally, combinations of different mechanisms of resistance may be synergistic. For example, introduction of NDM-5 alone or CirA deficiency alone in *K. pneumoniae* resulted in eightfold (0.5→4 mg/L) and twofold (0.5→1 mg/L) higher cefiderocol MIC. However, introduction of both NDM-5 and CirA deficiency resulted in a cefiderocol MIC > 256 mg/L [64].

### 2.5. Heteroresistance (In Vitro Emergence of Resistant Subpopulations)

Heteroresistance was systematically assessed in only two studies [20,33], both conducted by the same group. Heteroresistance was defined as presence of resistant (×2–4 the relevant CLSI breakpoints) subpopulations at a frequency of ≥1 in 10^6^ [20,33]. The prevalence of heteroresistance was much higher than that of nonsusceptibility, at 59% (64/108) in carbapenem-resistant *A. baumannii* (vs. 8.3% nonsusceptible), 48% (14/29) in *S. maltophilia* (vs. 0% nonsusceptible), 30% (27/89) in carbapenem-resistant *K. pneumoniae* (vs. 5.6% nonsusceptible), and 9% (6/69) in carbapenem-resistant *P. aeruginosa* (vs. 0% nonsusceptible). Furthermore, prevalence of heteroresistance was higher in carbapenem-resistant than in carbapenem-susceptible isolates and much lower in isolates susceptible to both carbapenems and extended spectrum cephalosporins [33], an observation that further supports the potential role of β-lactamases in facilitating emergence of cefiderocol resistance.

The frequency of cefiderocol-resistant subpopulations depends on a variety of factors, including methodological factors (such as bacterial density used considering potential inoculum effect [40,41]) as well as strain-specific factors [20,31,33,46,67]. With regard to the latter, and considering that cefiderocol resistance is a result of various different mechanisms acting in concert, a high baseline MIC appears to facilitate the in vitro and in vivo emergence of resistant subpopulations (i.e., the frequency of resistant subpopulations is higher) [40,57]. Similarly to what has been previously described for colistin heteroresistance [68,69], the frequency of resistant subpopulations increased after exposure to cefiderocol and decreased after removal from the drug [20].

### 2.6. In Vivo Emergence of Resistance or Reduced Cefiderocol Susceptibility

In vivo emergence of resistance was systematically addressed in two randomized controlled trials (CREDIBLE-CR [23] and APEKS-NP [22]) [21]. In these trials, a more than fourfold cefiderocol MIC increase during or after treatment was found in 7% (19 of 265) of the patients [21]. However, for most patients, the post-treatment cefiderocol MICs were relatively low. Specifically, post-treatment MICs were higher than susceptibility breakpoints in six (1.6%), three (1.1%), and four (1.5%) cases considering FDA, CLSI, and EUCAST breakpoints, respectively.

In vivo emerging resistance has also been reported in small observational case series [26,34,35,36]. In one study, 17 patients were treated with cefiderocol for difficult-to-treat *P. aeruginosa* infection, and relapses occurred in 3 of the patients, but repeat MIC testing was available in only 1 patient (fourfold increase from 0.25 to 1 mg/L) [26]. In another small series of 10 patients (*A. baumannii* n = 8, *K. pneumoniae* n = 2) who received cefiderocol as salvage therapy, microbiological failure was reported in 2 of the patients, but repeat MIC testing was available in only 1 (16-fold increase from 0.25 to 4 mg/L) [35]. In 13 patients with *A. baumannii* infections, microbiological failure was observed in about half (n = 7) of the patients but was associated with suboptimal attainment of PK/PD targets rather than emergence of resistance [36]. Finally, in 47 patients with carbapenem-resistant *A. baumannii* infections, microbiological failure occurred in 8 patients, in 4 of whom emergence of cefiderocol resistance was documented (MICs ranging from 4 to >32 mg/L) [34].

In addition, several case reports have described in vivo emergence of cefiderocol resistance: (1) on day 21 of treatment of a patient with hepatic abscesses by NDM-1 producing carbapenem-resistant *E. cloacae* (MIC 4→256 mg/L associated with *cirA* mutations) [24]; (2) on day 19 of treatment of a patient with intraabdominal abscesses by NDM-5-producing *E. coli* (MIC 2→>32 mg/L, associated with increased copy numbers of *bla_NDM_* genes) [25]; (3) after 32 days of a 6-week course of cefiderocol in a patient with pancreatic abscess by *P. aeruginosa* (a cefiderocol resistant *C. koseri* was also detected later in the same patient) [38]; (4) after a 3-week course of cefiderocol therapy in a patient with empyema by XDR *P. aeruginosa* (MIC 0.25→1 mg/L) [39]. In three of these four reports, emergence of resistance was associated with persistent or recurrent infection [24,25,38]. Notably, these reports involved patients with infections characterized by high bacterial load, insufficient source control, and/or prolonged treatment durations, which provide ideal conditions for in vivo emergence of resistance. Indeed, a higher bacterial density increases the chance of emergence of resistant subpopulations [40]. Furthermore, an inoculum effect has been described by which higher cefiderocol MIC is observed in/associated with higher bacterial densities [41].

In vivo emergence of cross-resistance to cefiderocol in ceftazidime/avibactam-treated patients has also been described [14,42]. Bianco et al. [14] described in vivo emergence of cross-resistance between cefiderocol and ceftazidime/avibactam after ceftazidime/avibactam treatment in 16 patients with KPC-producing *K. pneumoniae* isolates. Similarly, Tiseo et al. [42] reported emergence of cross-resistance to cefiderocol during treatment with ceftazidime/avibactam resulting from emergence of a KPC-3 variant (KPC-31). In addition, emergence of cross-resistance to cefepime, ceftazidime/avibactam, and cefiderocol in *Enterobacter hormaechei* strains resulting from emergence of A292_L293del AmpC variant was described in two cefepime-treated patients [43].

Finally, the potential of in vivo emergence of cross-resistance between ceftolozane/ tazobactam and cefiderocol in *P. aeruginosa* has been reported [16,44]. Analysing 16 pairs (before/after ceftolozane/tazobactam treatment) of *P. aeruginosa* isolates Simner et al. detected ≥4-fold increases in cefiderocol MICs in 4 of the 16 isolates, although the MIC was above CLSI susceptibility breakpoint in only 1 case [16]. In a case report, emergence of cross-resistance to cefiderocol was reported in a ceftolozane/tazobactam-treated patient [44]. Potential contributing mechanisms involved an amino acid substitution in *AmpC* [16,44], overexpression of the MexAB–OprM efflux pump [16], and mutations in siderophore receptors PiuD and PirA [44], although the relative contribution of each of these mechanisms to cefiderocol resistance was unclear from these studies [16,44].

## 3. Discussion

### 3.1. Overview of Mechanisms of Cefiderocol Resistance

Resistance to cefiderocol is associated with a combination of mechanisms, each contributing to reduced cefiderocol susceptibility; β-lactamases, mutations affecting expression/function of siderophore receptors (most commonly involved genes: *cirA* and *fiu* in Enterobacterales, *piuA* and *piuD* in *P. aeruginosa, pirA* and *piuA* in *A. baumannii*) and to a lesser extent mutations affecting expression/function of porins and/or efflux pumps, or target (PBP-3) modification. Each of these mechanisms alone are usually insufficient to raise cefiderocol MIC above PK/PD breakpoints. Therefore, cefiderocol resistance is typically the result of various combinations of the aforementioned mechanisms of resistance. A combination of β-lactamases and reduced permeability (due to mutations affecting porins or siderophore receptors) appears to be the most common mechanism resulting in cefiderocol nonsusceptibility.

### 3.2. Role of β-Lactamases in Cefiderocol Resistance

One of the main advantages of cefiderocol is that it remains active (MIC below susceptibility breakpoints) against the majority of isolates producing various β-lactamases, including isolates that are resistant to carbapenems, ceftazidime/avibactam, ceftolozane/tazobactam, meropenem/vaborbactam, and imipenem/relebactam [8,9,10,11,12,13]. However, cefiderocol is not completely stable against various β-lactamases. Reduced susceptibility has been reported in the presence of specific β-lactamases, including KPC variants conferring resistance to ceftazidime/avibactam, AmpC variants conferring resistance to ceftazidime/avibactam and/or ceftolozane tazobactam, selected SHV- and PER-type ESBLs, selected NDM, and OXA-427. Of interest is that cefiderocol resistance mediated by serine β-lactamases can be reversed by avibactam [19,29,47], which could be a useful combination in clinical practice to overcome cefiderocol resistance. Similarly, MBL inhibitors (when they become available) could prove useful in combination with cefiderocol against NDM-producing isolates.

Generally, β-lactamases alone are not sufficient to raise cefiderocol MIC above susceptibility breakpoints, and nonsusceptibility is usually the result of coexpression of multiple β-lactamases and/or overexpression of β-lactamases, possibly in combination with other mechanisms of resistance described above (especially mutations associated with reduced permeability). Additionally, β-lactamases can facilitate the emergence of resistance by additional mechanisms [33,40]. Notably, prevalence of cefiderocol resistance was very high is some cohorts of isolates producing specific β-lactamases (OXA-427 [15], KPC variants conferring resistance to ceftazidime/avibactam [14], NDM [12,19,62]). This suggests that emerging β-lactamases could become a major contributor to cefiderocol resistance in the future.

### 3.3. Cross-Resistance between Other Antibiotics and Cefiderocol

Cefiderocol appears to be active against the vast majority of Gram-negative bacteria that are nonsusceptible to ceftazidime/avibactam or ceftolozane/tazobactam [8]. However, cross-resistance among ceftazidime/avibactam, ceftolozane/tazobactam, and cefiderocol has been reported, associated with KPC variants in *K. pneumoniae* [14,41,55] or AmpC variants in *Enterobacter* spp. [43] and *P. aeruginosa* [16]. Notably, in a recent study, cefiderocol resistance was very high (83%) in ceftazidime/avibactam-resistant KPC-producing Enterobacterales (predominantly *K. pneumoniae*) [14]. Furthermore, metallo-β-lactamases, which are known to confer resistance to ceftazidime/avibactam and ceftolozane/tazobactam, have also been associated with decreased susceptibility to cefiderocol (especially NDM metallo-β-lactamases) [12,25,29,40,54]. Therefore, widespread clinical use of ceftazidime/avibactam and ceftolozane/tazobactam may contribute to the emergence and spread of cefiderocol resistance, even in the absence of exposure to cefiderocol.

On the other hand, cross-resistance to cefiderocol and ceftazidime/avibactam conferred by KPC variants may be associated with reversal of susceptibility to carbapenems [14,42] and may be amenable to treatment with meropenem/vaborbactam [42]. Furthermore, *P. aeruginosa* AmpC variants conferring resistance to cefiderocol, ceftazidime/avibactam, and ceftolozane/tazobactam may remain susceptible to imipenem/relebactam [16].

Finally, potential for cross-resistance between colistin and cefiderocol has been proposed [37], although cefiderocol appears to retain activity against the majority of colistin-resistant Enterobacterales [12,61] and colistin-resistant *A. baumannii* [63].

### 3.4. Importance of Heteroresistance and In Vivo Emergence of Resistance

The clinical impact of heteroresistance is that resistant subpopulations can emerge during treatment, resulting in treatment failure and the spread of resistant strains [68,69,70]. The high prevalence of heteroresistance to cefiderocol has been proposed as an explanation for the suboptimal efficacy of cefiderocol against carbapenem-resistant bacteria, especially *A. baumannii* [20,33]. However, there are no clinical data to support this hypothesis, and emergence in vivo of resistance to cefiderocol during treatment appears to be rare (<2%) in clinical studies [21,71].

The potential impact of cefiderocol heteroresistance in the longer term and as cefiderocol is increasingly being used in clinical practice is yet unclear. Yearly data (2014–2019) from the SIDERO-WT studies are not informative, considering that cefiderocol was approved only in 2020 in North America and Europe and remains unavailable in some countries. Based on experience with polymyxins (with a similarly high prevalence of heteroresistance [68] and similarly low reported frequency of in vivo emerging resistance [69]), emergence of resistance in cefiderocol is likely in the future, especially in *A. baumannii*.

Notably, the definition of and methodologies for detecting heteroresistance are not well-defined [68,70,72,73]. Furthermore, traditional susceptibility testing methods, which are based on a few individual colonies (rather than the bacterial population), may miss resistant subpopulations even if they are present at a relatively high frequency [73]. Additionally, heteroresistance can be unstable and very dynamic (resistant subpopulations rapidly increasing/decreasing in the presence/absence of antibiotic pressure), which may also result in failure to detect resistance (reversal to the susceptible phenotype has been documented after only 12 h of growth in blood culture flasks) [74]. Finally, appropriately designed clinical studies are needed to assess the impact of heteroresistance on clinical outcomes and to uncover characteristics of heteroresistance (in combination with patient characteristics) that predict treatment failure [70]. Such studies are necessary to define appropriate methods to detect resistance (and clinically relevant heteroresistance).

In theory, the following factors (often acting in concert) can facilitate clinically relevant in vivo emergence of resistance [69,70,73]: (1) infections associated with higher bacterial burden or poor source control, (2) higher baseline MIC (closer to breakpoints of resistance), (3) higher frequency of resistant subpopulations and especially of resistant subpopulations with preserved fitness and virulence, (4) immunosuppression, (5) monotherapy, and (6) failure to achieve appropriate PK/PD targets at the site of infection.

### 3.5. Limitations

Mutations identified by in vitro selection of mutants following exposure to cefiderocol may not be relevant to in vivo observed mutations, as has been described previously for colistin [68,69]. Mutations observed in vitro, especially those affecting iron transport [58,60], may be associated with fitness cost [40,58] and can be unstable or reversible in the absence of continued cefiderocol exposure [58,60]. This is further supported by the fact that resistance emerging in vivo in animal models [58,67] or clinical studies [21,22,23,75] is much less frequent than heteroresistance [20] and much less frequent than resistance emerging in vitro after exposure to cefiderocol. Furthermore, mutations identified in cefiderocol-resistant clinical isolates do not prove causality. Additionally, mutations identified in single-centre cohorts may not be generalizable to other sites and may overestimate the role of certain mechanisms of resistance by considering only isolates that represent one or few related clones. Nevertheless, comparisons of in vivo emerging cefiderocol resistant strains with their parental strains are very useful for identifying clinically relevant mechanisms of resistance [21,22,23,75]. Furthermore, on many occasions, the role of various resistance mechanisms has been confirmed in isogenic mutants.

Finally, it has recently been suggested that in vitro susceptibility testing of cefiderocol against *A. baumannii* may overestimate its activity compared with conditions in vivo [76]. Specifically, cefiderocol MIC was 2- to 16-fold higher in the presence of human serum, human serum albumin, or human pleural fluid in the culture medium [76]. Under these conditions, higher expression of β-lactam resistance-associated genes (β-lactamases and PBPs) was observed, combined with downregulation of iron uptake-related genes, which could explain the higher cefiderocol MIC [76], and may explain lower efficacy of cefiderocol against carbapenem-resistant *A. baumannii* infections [23].

## 4. Materials and Methods

### 4.1. Search Strategy and Sources

PubMed and Scopus were searched from inception to 6 May 2022, using the keywords cefiderocol OR s-649266. After deduplication, screening for eligibility of retrieved articles was conducted by the first author using the Rayyan online app for collaborative systematic reviews [77]. Eligibility of included articles was validated by a second author.

### 4.2. Eligibility Criteria

The following types of studies were eligible for review: (1) cohorts of cefiderocol-resistant clinical isolates evaluating for the presence of specific resistance mechanisms in these isolates (e.g., by PCR or whole genome sequencing to identify mutations in genes potentially contributing to cefiderocol nonsusceptibility); (2) cohorts of clinical isolates reporting high proportions (>20%) of cefiderocol resistance and correlating such resistance with specific mechanisms; (3) studies trying to identify resistance mechanisms by comparing in vivo emerging cefiderocol-resistant clinical isolates with their parental strains; (4) studies trying to identify resistance mechanisms by comparing in vitro derived (by subculturing in the presence of cefiderocol) cefiderocol-resistant isolates with their parental strains; (5) studies evaluating the impact of specific resistance mechanisms on cefiderocol minimum inhibitory concentration (MIC) by introducing the relevant resistance determinants in vitro (comparison of the cefiderocol MIC of the derived with that of the isogenic parental strain); (6) studies examining the prevalence of heteroresistance; (7) studies evaluating for the emergence of resistance in vivo during or after cefiderocol treatment (even if no evaluation for the mechanism of resistance was conducted).

The following types of studies were excluded: (1) nonoriginal articles (e.g., reviews, commentaries, editorials); (2) irrelevant original articles (not satisfying the eligibility criteria described in the paragraph above); (3) potentially relevant articles written in languages other than English.

### 4.3. Data Items

The following data were extracted from each eligible article: mechanisms of resistance, methods for confirming contribution of each mechanism to cefiderocol resistance, proportion of heteroresistance and the definition thereof used, and frequency of emergent resistance during treatment in clinical studies (case reports of emergent resistance were also recorded).

### 4.4. Assessment of the Evidence for the Reported Mechanisms of Resistance

Studies were grouped (groups 1 to 6) as described above in Section 4.2. The evidence for a resistance mechanism was considered strongest if both of the following were true: (1) the resistance mechanism was detected in cefiderocol-resistant clinical isolates (group 1, 2, or 3 studies) and (2) resistance determinants detected in clinical isolates were confirmed to be able to raise cefiderocol MIC in vitro (group 5 studies). Detection of resistance mechanisms in in vitro derived cefiderocol-resistant isolates (group 4 studies) in the absence of detection of the same mechanisms in vivo suggests that these mechanisms may not be clinically relevant (e.g., due to fitness cost in vivo [69]). Similarly, confirmation in vitro that specific resistance determinants can raise cefiderocol MIC (group 5 studies) does not necessarily mean that these mechanisms are relevant/common in vivo. Finally, detection of potential resistance determinants in clinical isolates does not, alone, confirm that the detected mechanism truly contributes to cefiderocol nonsusceptibility. More than one mutation in potentially relevant resistance genes may be present simultaneously, and the relative (if any) contribution to cefiderocol nonsusceptibility of each mutation would be unclear in group 1–2 studies (and to a lesser extent in group 3 studies).

### 4.5. Synthesis of Results

A qualitative synthesis of the data was conducted. The various potential resistance mechanisms were recorded, and the relevant evidence supporting each reported mechanism was assessed as described above.

## 5. Conclusions

Although cefiderocol appears to retain activity against most XDR Gram-negative bacteria, resistance is increasingly being reported and is high in some cohorts. Various mechanisms of resistance have been identified, including β-lactamases, mutations affecting siderophore receptors, mutations affecting porins and efflux pumps, and mutations in PBP-3 (the target of cefiderocol). However, on many occasions, the mechanism of resistance remains unclear or appears to result from a combination of mechanisms. Especially worrisome are the emergence of various β-lactamases able to cause multifold increases in cefiderocol MIC and the high prevalence of cefiderocol resistance in the presence of selected β-lactamases (mainly NDM-1, KPC-variants conferring resistance to ceftazidime/avibactam, and OXA-427). Heteroresistance is also highly prevalent, mainly in *A. baumannii*, but its clinical impact is yet unclear, and emergence of resistance during treatment is uncommon based on available data. Continued surveillance of cefiderocol activity is important as this agent is being introduced in clinical practice.

## Figures and Tables

**Figure 1 antibiotics-11-00723-f001:**
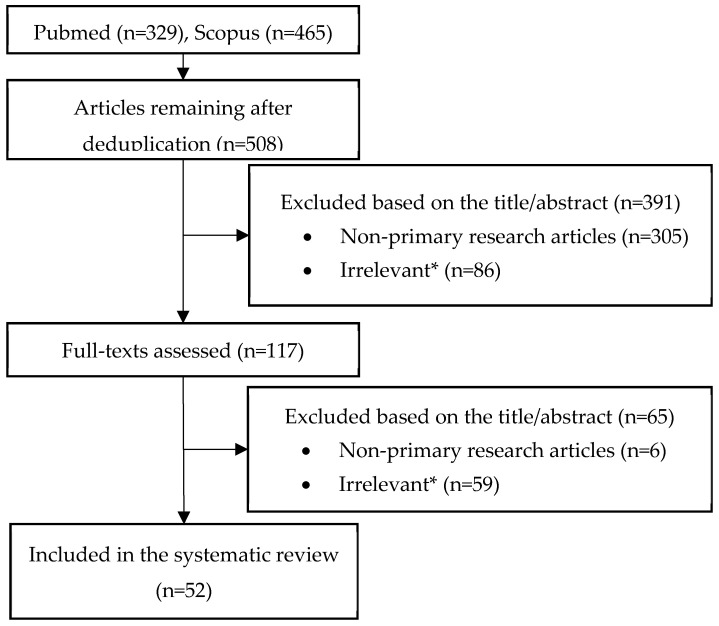
Flow chart. * not satisfying the eligibility criteria described in Methods.

## Data Availability

Data analyzed in this systematic review were a re-analysis of existing data from the literature and are available in the text and tables of the article and from sources cited in the reference section.

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
