# Peer review of "Cefiderocol: Systematic Review of Mechanisms of Resistance, Heteroresistance and In Vivo Emergence of Resistance"

_antibiotics, 2022, doi:10.3390/antibiotics11060723_

Round 1
Reviewer 1 Report
This systematic review provides a comprehensive analysis of relevant literature in order to describe mechanisms of resistance to cefiderocol as well as prevalence of heteroresistance and in-vivo emerging resistance. Overall, it is well written but authors should improve the introduction. More specifically they should provide information regarding the structure of cefiderocol and they should explain for unfamiliar readers what are siderophores , what are the major siderophores (and siderophore receptors) in pathogens such as P. aeruginosa or Enterobacteriales f.i. Furthermore, authors should correct the email address of corresponding author.
Author Response
We thank the reviewer. Please find a point-by-point reponse below:
- “This systematic review provides a comprehensive analysis of relevant literature in order to describe mechanisms of resistance to cefiderocol as well as prevalence of heteroresistance and in-vivo emerging resistance. Overall, it is well written but authors should improve the introduction. More specifically they should provide information regarding the structure of cefiderocol and they should explain for unfamiliar readers what are siderophores , what are the major siderophores (and siderophore receptors) in pathogens such as P. aeruginosa or Enterobacteriales f.i.”
- Reply: We thank the reviewer. We agree that an Introduction on siderophores is necessary for the unfamiliar readers. A relevant paragraph was added in the Introduction;
- “Cefiderocol is a novel siderophore cephalosporin [6,7]. It has a structure similar to cefepime (pyrrolidinium group on the C-3 side chain, which improves stability against β-lactamases) and ceftazidime (same C-7 side chain conferring improved stability against β-lactamases and improved transport across the outer membrane) [6,7]. The major difference is the addition of a chlorocatechol goup on the end of the C-3 side chain which further enhances β-lactamase stability and confers siderophore activity [6,7]. Siderophores are natural iron-chelating molecules used by bacteria to facilitate iron transport. Of note, the natural siderophores enterobactin (E. coli) and pyoverdine (P. aeruginosa) contain similar catechol groups as an iron chelating moiety. By utilizing natural iron transportation systems (often referred to as a “Trojan horse” strategy) cefiderocol is actively transported across the outer membrane of Gram-negative bacteria therefore overcoming resistance mediated by porin loss or efflux pumps [6]. This in combination with improved β-lactamase stability makes cefiderocol an ideal candidate for treatment of infections by XDR/PDR Gram-negative bacteria.”
- Reply: We thank the reviewer. We agree that an Introduction on siderophores is necessary for the unfamiliar readers. A relevant paragraph was added in the Introduction;
- “Furthermore, authors should correct the email address of corresponding author.”
- Reply: Thank you for pointing the mistake. The first author is the corresponding author. The email address is correct (it is the asterisk that was misplaced)
Reviewer 2 Report
Bravo!
A very systematic overview of Cefiderocol resistance and heteroresistance. Valuable information for the wider community and deserves follow-up in the future.
I would insist that the authors add a paragraph stating that the methodology of detecting resistance (and heteroresistance) needs an urgent update. Common antibiotic susceptibility test strongly underestimates the occurrence of (hetero)resistance in clinical isolates. Please use examples from literature (Pereira et al, 2021; https://www.nature.com/articles/s42003-021-02052-x) and (Brukner and Oughton, 2020; https://www.ncbi.nlm.nih.gov/pmc/articles/PMC7461948/).
This is actually increasing the value and importance of the authors’ message.
Please define better “irrelevant” on Tab 1 or exclude them.
Author Response
We thank the reviewer. Please find below a point-by-pont response:
- “I would insist that the authors add a paragraph stating that the methodology of detecting resistance (and heteroresistance) needs an urgent update. Common antibiotic susceptibility test strongly underestimates the occurrence of (hetero)resistance in clinical isolates. Please use examples from literature (Pereira et al, 2021; https://www.nature.com/articles/s42003-021-02052-x) and (Brukner and Oughton, 2020; https://www.ncbi.nlm.nih.gov/pmc/articles/PMC7461948/).”
- Reply: We thank the reviewer for the suggestion and for the proposed references (both very useful). We agree that consensus is needed to define heteroreristance as well as the methodology to detect resistance and heteroresistance. Furthermore, clinical studies are needed to assess the impact of heteroresistance on clinical outcomes or emergence of resistance. The following paragraphs were added in Discussion, section 3.4:
- “Of note, definition of and methodology for detecting heteroresistance are not well-defined [68,70,72,73]. Furthermore, traditional susceptibility testing methods, which are based on few individual colonies (rather than the bacterial population), may miss resistant subpopulation even if present at a relatively high frequency [73]. Additionally, heteroresistance can be unstable and very dynamic (resistant subpopulations rapidly increasing/decreasing in the presence/absence of antibiotic pressure) which may also result in failure to detect resistance (reversal to the susceptible phenotype has been documented after only 12 hours of growth in blood culture flasks) [74]. Finally, appropriately designed clinical studies are needed to assess the impact of heteroresistance on clinical outcomes and to uncover characteristics of heteroresistance (in combination with patient characteristics) that predict treatment failure [70]. Such studies are necessary to define appropriate methods to detect resistance (and clinically relevant heteroresistance).”
- “In theory the following factors (often acting in concert) can facilitate clinically relevant in vivo emergence of resistance [69,70,73]: (1) infections associated with higher bacterial burden or poor source control, (2) higher baseline MIC (closer to breakpoints of resistance), (3) higher frequency of resistant subpopulations and especially of resistant subpopulations with preserved fitness and virulence, (4) immunosuppression, (5) mono-therapy, (6) failure to achieve appropriate PK/PD targets at the site of infection.”
- Reply: We thank the reviewer for the suggestion and for the proposed references (both very useful). We agree that consensus is needed to define heteroreristance as well as the methodology to detect resistance and heteroresistance. Furthermore, clinical studies are needed to assess the impact of heteroresistance on clinical outcomes or emergence of resistance. The following paragraphs were added in Discussion, section 3.4:
- “Please define better “irrelevant” on Tab 1 or exclude them.”
- Reply: We assume the reviewer is referring to Figure 1. As explained in Methods (section 4.2) irrelevant articles were defined as “irrelevant original articles (not satisfying the eligibility criteria described in the paragraph above)”. A legend was added below the Figure: “* not satisfying the eligibility criteria described in Methods”
Reviewer 3 Report
In this review, the authors summarized the current state of the field for resistance emerging to cefiderocol. The document is well-written and highly informative. The review is an interesting read. Mostly it appears that a combination of resistance mechanisms is necessary in order to confer high levels of cefiderocol resistance.
Minor comments:
1. There are a few instances where Genus and species of the bacteria are not italicized.
2. There are a few instances were gene and proteins names are mixed. Genes should be lowercase italicized (cirA or blaNDM). Proteins should be capitalized (CirA or NDM).
3. What type of OXA is OXA-427, OXA-48 like - it is one of those missing part of the b5-b5 beta-strand that is ceftazidime resistant? Please clarify.
4. Some of the beta-lactamases listed in the Table are not mentioned at all at the text. Suggest adding a single sentence or two.
5. Lines 212 and 333, spelling "carbapenems"
6. Line 314, spelling "ceftazidime"
7. Cefiderocol is mostly written out, but then abbreviated in Table 3. Please change for consistency.
Author Response
We thank the reviewer. Please find below a point-by-point response:
- “There are a few instances where Genus and species of the bacteria are not italicized.”
- Reply: Thank you. The manuscript was re-checked and appropriate corrections were made
- “There are a few instances were gene and proteins names are mixed. Genes should be lowercase italicized (cirA or blaNDM). Proteins should be capitalized (CirA or NDM).”
- Reply: Thank you. The manuscript was re-checked and appropriate corrections were made
- “What type of OXA is OXA-427, OXA-48 like - it is one of those missing part of the b5-b5 beta-strand that is ceftazidime resistant? Please clarify.”
- Reply: OXA-427 is not similar to OXA-48. It indeed confers resistance to extended-spectrum cephalosporins (mostly ceftazidime) The following text (and relevant reference) was added:
- “OXA-427 is a novel type of carbapenem-hydrolysing class D β-lactamase, sharing only 22%-29% amino acid identity with OXA-48-like enzymes and conferring resistance to extended-spectrum cephalosporins (mostly ceftazidime) [65]”
- Reply: OXA-427 is not similar to OXA-48. It indeed confers resistance to extended-spectrum cephalosporins (mostly ceftazidime) The following text (and relevant reference) was added:
- “Some of the beta-lactamases listed in the Table are not mentioned at all at the text. Suggest adding a single sentence or two.”
- Reply: We believe the flow of the manuscript would become tiring for readers if all relevant β-lactamases were mentioned in the main text. Instead, we chose to focus the main text on β-lactamases whose role on cefiderocol resistance is more established.
- “Lines 212 and 333, spelling "carbapenems"”
- Reply: Corrected
- “Line 314, spelling "ceftazidime”
- Reply: Corrected
- “Cefiderocol is mostly written out, but then abbreviated in Table 3. Please change for consistency”
- Reply: CFDC abbreviation was removed